# Lived Experiences of Recovery from Severe Depression with Psychotic Symptoms and Suicidal Behaviors: A Phenomenological Study

**DOI:** 10.3390/ijerph22111606

**Published:** 2025-10-22

**Authors:** Saifon Aekwarangkoon, Earlise Ward, Sirintra Duangsai, Sangtien Jearawattanakul

**Affiliations:** 1Division of Psychiatric and Mental Health Nursing, School of Nursing, Excellence Center of Community Health Promotion, Walailak University, Nakhon Si Thammarat 80160, Thailand; sirintra.du1509@gmail.com (S.D.); sangtien@gmil.com (S.J.); 2School of Medicine and Public Health, University of Wisconsin-Madison, 610 N. Whitney Way, Madison, WI 53706, USA; earlise.ward@fammed.wisc.edu

**Keywords:** recovery, suicidal behaviors, depression, psychotic symptoms, psychological resilience, mental health, patient experience

## Abstract

Severe depression with psychotic symptoms and suicidal behaviors is a critical mental health condition requiring comprehensive care. While clinical interventions are necessary, less is known about the lived experiences of individuals who recover from such complex states. This study explores the lived experiences of recovery among individuals diagnosed with major depressive disorder with severe depression, psychotic symptoms, and suicidal ideation or suicide attempts, focusing on how they found meaning in their journey and maintained recovery over time. A phenomenological approach was employed. In-depth interviews were conducted with nine individuals who had experienced severe depression with psychotic symptoms and suicidal behaviors, received psychiatric treatment, and later achieved recovery. Data were analyzed using interpretive phenomenological analysis. Participants described recovery as a deeply personal and transformative journey. Three core themes emerged: (1) understanding and reframing internal experiences, (2) drawing strength from therapy, relationships, and self-care, and (3) gradually regaining agency, identity, and meaning in life. Recovery from severe depression with psychotic symptoms and suicidal behaviors is possible. Mental health nurses and professionals play a vital role in supporting this journey through person-centered, holistic, and empowering care approaches.

## 1. Introduction

Suicidal behaviors, depression, and psychotic symptoms are pressing mental health challenges with rising prevalence and complexity. These often interrelated conditions contribute significantly to the global disease burden. According to the World Health Organization [1], over 280 million people live with depression, and nearly 800,000 die by suicide each year. When depression includes psychotic features such as hallucinations or delusions, the illness becomes more severe, harder to treat, and significantly increases the risk of self-harm [2]. In Asia, particularly in Japan, South Korea, and Thailand, suicide has become a growing public health concern, especially among adolescents and older adults [3]. In Thailand, suicide rates have increased by 15 percent over the past decade, a trend closely linked to the rising prevalence of depression and psychotic symptoms, as evidenced by a steady increase in psychiatric service utilization [4]. These developments highlight a deepening mental health crisis and underscore the urgent need for early intervention, public awareness, and accessible, integrated mental health care.

Suicidal behaviors, depression, and psychotic symptoms not only cause profound suffering for those affected but also severely impact families, healthcare systems, and society as a whole. Individuals often endure persistent hopelessness, social withdrawal, and functional decline [2]. Families face emotional and financial strain, with caregiving demands contributing to mental health challenges and breakdowns in relationships [3]. Healthcare systems are burdened by frequent relapses and long-term treatment needs, leading to rising costs and resource shortages [4]. At the societal level, stigma and exclusion limit recovery and social reintegration, resulting in lost potential and weakened social cohesion [1]. Among youth and working-age adults, these interconnected issues carry serious socioeconomic consequences and threaten national development [5]. These impacts underscore the urgent need for coordinated, compassionate, and systemic mental health care responses.

Living with suicidality, severe depression, and psychotic symptoms is an exhausting experience shaped by complex interactions of genetic vulnerability, trauma, and psychosocial stress [6,7]. Many individuals engage repeatedly in treatment and seek social support, yet the combined impact of emotional distress, psychotic symptoms, and treatment effects often undermines their resilience [8]. When progress is slow or uncertain, fatigue and withdrawal naturally emerge as coping responses. This cycle of isolation can deepen feelings of shame and hopelessness, which in turn may intensify suicidal thoughts and psychotic experiences [9]. These persistent challenges highlight the importance of compassionate care that acknowledges the lived realities of chronic mental health struggles and calls for deeper understanding and comprehensive approaches beyond symptom management alone [10,11].

Effective recovery from depression, suicide, and psychotic symptoms requires a comprehensive approach that integrates both internal and external sources of healing. On the intrapsychic level, developing self-awareness enables individuals to recognize emotional triggers and manage symptoms [12,13], while psychological therapies help address emotional challenges, strengthen coping skills, and foster resilience [14]. In addition, external supports play a critical role. Mental health professionals provide personalized guidance tailored to individual needs [15]; supportive environments such as calming nature, peaceful homes, and positive social interactions promote well-being [16]; and social support from family, friends, and peers helps reduce isolation and encourages treatment adherence [17]. Together, these internal and external elements form a strong foundation for sustained recovery and renewed purpose [18].

Despite the effectiveness of holistic recovery approaches, significant gaps remain in understanding and treating severe depression with psychotic symptoms and suicidal behaviors. A major gap is the lack of insight into patients’ lived experiences, which leads to a disconnect between clinical care and the real needs of patients. Without understanding the emotional and psychological struggles individuals face, treatments may fail to address their core challenges. This study aims to explore these lived experiences, providing crucial insights that will guide more personalized and compassionate care. By better understanding the recovery process, patients can gain clarity and empowerment, while healthcare providers and families can respond more effectively to their needs [19]. This research has the potential to improve patient outcomes, support families, and enhance the healthcare system, offering hope for change and better integration of mental health care [20].

### Mental Health Care in the Thai Context

Thailand is facing increasing challenges in caring for individuals with psychiatric disorders, including schizophrenia, bipolar disorder, and major depressive disorder. Among these, severe depression with psychotic symptoms and suicidal behaviors is especially complex and prevalent [21,22]. As a result, severe depression is the primary focus of our research. While most individuals remain in the community, only those in acute crisis are hospitalized for intensive care [21]. Depression, suicidal ideation, and psychotic symptoms are key concerns across diagnoses and contribute significantly to relapse, poor outcomes, and prolonged suffering for individuals, families, and the healthcare system [1].

After discharge, long-term care often falls to community nurses who typically lack formal psychiatric training and work under limited supervision from mental health professionals [22]. Their responsibilities include monitoring medication, managing side effects, and providing basic psychoeducation, such as advising families to reduce stress, avoid substance use, and watch for suicide risk, especially in depression [8]. However, the current system focuses mainly on symptom control and risk prevention, with limited access to recovery-oriented services like psychotherapy or emotional support, particularly in rural areas. Private care is available but often unaffordable [23]. These challenges point to the urgent need for integrated mental health care that addresses both medical and psychosocial needs at the community level.

## 2. Materials and Methods

Colaizzi’s descriptive phenomenological approach [24,25] was employed to explore the lived experiences of individuals diagnosed with major depressive disorder (MDD) according to the criteria outlined in the Diagnostic and Statistical Manual of Mental Disorders, Fifth Edition (DSM-5) [26], who had experienced suicidal behaviors and psychotic symptoms. All participants received ongoing psychiatric care and psychotherapy at the Smile Clinic, School of Nursing, Walailak University. The therapeutic protocol reflected current practice in Thailand, which emphasizes symptom stabilization and follow-up, rather than structured recovery-oriented interventions.

Purposive sampling was used with the following inclusion criteria: (a) aged 18–60 years; (b) sustained recovery with no recurrence of major symptoms for at least two years; (c) consistent follow-up with both a psychiatrist and a psychiatric nurse; and (d) willingness to participate in a qualitative interview. Exclusion criteria included (a) any condition that could impair the participant’s ability to engage in the interview process (e.g., severe cognitive impairment, active psychosis, or current substance intoxication); and (b) lack of informed consent. Nine participants (three identifying as LGBTQ+ and six female), aged 19 to 57 years, met the criteria. All had been diagnosed with MDD and had experienced both suicidal behaviors and psychotic symptoms (e.g., hallucinations, delusions). Most participants were Buddhist, with education ranging from high school to master’s degree. Six were single, and three were in relationships. Family dynamics ranged from distant to supportive. Occupations included students, government employees, an NGO worker, and a housewife. Financial situations varied.

Individual, in-depth interviews were conducted in a private, supportive environment using a flexible, open-ended guide designed to elicit reflective responses. Participants were invited to explore their lived experiences through six core questions: (1) Can you share your experience of going through depression, suicide, and psychosis? (2) What was the moment you first felt things starting to improve? What helped you notice that? (3) How did you realize recovery had begun? What helped you keep moving forward? (4) What kind of support or care helped you most in your healing process? (5) Looking back, how has recovery changed the way you see yourself and your life today? and (6) What challenges still remain for you in staying well or feeling fully recovered?

Each interview lasted approximately 60 to 90 min. The primary interviewer (SA), a specialist in psychiatric nursing, facilitated the conversations. A second researcher (SD), also trained in psychiatric nursing, observed nonverbal behavior and took detailed field notes. This dual-researcher method enhanced the credibility and depth of the data. To build rapport, sessions began with informal conversation before moving to the main interview. Verbal responses were complemented by observations of tone, body language, and facial expression. Data collection and preliminary analysis were conducted concurrently, allowing for real-time adjustment of the interview process and early identification of emerging themes. Data collection continued until saturation was reached, indicating no new information was emerging.

Data were analyzed using Colaizzi’s seven-step method to ensure a structured and rigorous approach to theme development: (1) Participants’ responses were transcribed verbatim, with accuracy verified through repeated listening and careful reading; (2) Significant statements related to the research phenomenon were identified; (3) Formulated meanings were derived through reflective analysis of these statements and participants’ experiences; (4) These meanings were grouped into themes according to shared characteristics; (5) A comprehensive description of the phenomenon was developed by integrating all themes; (6) Reflexive discussions were conducted among researchers to acknowledge and minimize potential biases; and (7) Findings were validated through peer debriefing and member checking, with discrepancies resolved through consensus. All analysis was conducted manually. Throughout the process, the research team remained reflexive, critically examining their assumptions to reduce subjectivity and strengthen analytical rigor.

Trustworthiness was ensured through data triangulation by systematically comparing interview transcripts with detailed field notes to confirm consistency and depth of information. Peer review was conducted by three independent qualitative researchers who critically examined coding processes, theme development, and interpretations to reduce potential bias. An audit trail documented all analytic decisions and procedures to ensure transparency and confirm data relevance. Maximum variation sampling was operationalized by intentionally selecting participants with diverse characteristics, including age, gender identity, religion, education level, family dynamics, and occupation, to capture a broad spectrum of experiences related to depression, suicidal behavior, and psychotic symptoms, thereby enhancing the study’s confirmability, credibility, and transferability.

The research protocol was approved by the Committee on Human Rights Related to Research Involving Human Subjects, Walailak University, Thailand (Approval No. WUEC-25-208-01). Informed consent was obtained from all participants for the use of de-identified secondary data, including psychotherapy progress notes and interview transcripts, in accordance with ethical research principles.

## 3. Results

The findings highlight the lived experiences of participants as they overcame depression, suicidal tendencies, and psychiatric symptoms, revealing three main themes: (1) understanding and reframing internal experiences, (2) drawing strength from therapy, relationships, and self-care, and (3) gradually regaining agency, identity, and meaning in life. Representative quotations were purposively selected to illustrate these themes, reflecting common patterns across all nine participants. While not every participant explicitly discussed every sub-theme, all themes were broadly shared, with Theme 2 emerging as the most salient, frequently described as a pivotal source of psychological strength. Themes 1 and 3 were also prominent but varied in emphasis among individuals. A comprehensive summary of themes and sub-themes is presented in Table 1.

### 3.1. Understanding and Reframing Internal Experiences

This theme focuses on two key processes: making sense of trauma as a step toward healing, and reconstructing self-narratives with compassion. By facing and understanding their trauma, individuals begin to transform their pain into a path to recovery. Rebuilding self-narratives with kindness allows them to shift from self-blame to self-acceptance, recognizing their resilience and growth.

#### 3.1.1. Making Sense of Trauma as a Step Toward Healing

Understanding the depth of emotional pain became a turning point. By facing the trauma, participants began to make sense of what once felt unbearable, opening the door to healing.


*“…I felt myself falling into a dark hole, filled with pain that wouldn’t go away. I couldn’t tell where I ended and the pain began. My thoughts and feelings were too much. I couldn’t breathe. There was no way out, only pain and the thought of death. But when I truly felt how deep the pain was, something in me started to look for a way to keep going…”*
(P6)


*“…For a long time, I tried to forget everything. I thought ignoring it would make it disappear, but it only made the memories stronger. When I finally admitted to myself that I had been hurt, I could begin to understand why I felt so broken—and that I didn’t deserve to suffer in silence…”*
(P2)

#### 3.1.2. Reconstructing Self-Narratives with Compassion

Rebuilding self-narratives with compassion allows individuals to shift from seeing themselves through the lens of pain and failure to embracing self-acceptance and understanding. Recognizing their efforts and growth, they learn to heal with kindness and self-compassion.


*“…I used to see myself as broken and weak, like my story was only about failure. But when I stopped asking what was wrong with me and started asking what I needed, something shifted. I began to see someone who was trying, surviving, and worthy of kindness—not a problem to be fixed, but a person to be cared for…”*
(P2)


*“…I always thought I was the problem—that I was weak, lazy, or just too broken. But in therapy, I started to look at myself differently. I began to see someone who had been trying to survive impossible things. That changed everything. I wasn’t just a diagnosis anymore; I was a human being with a story that mattered…”*
(P7)

### 3.2. Drawing Strength from Therapy, Relationships, and Self-Care

This theme illustrates how healing is supported through meaningful therapeutic processes, compassionate relationships, and gentle self-reconnection. Participants described therapy as a path to rediscovering themselves, relationships as sources of deep support and understanding, and self-care as a return to inner wholeness. Each aspect offers strength to face pain and trust in the possibility of change.

#### 3.2.1. Therapy as a Journey of Self-Discovery, Healing, and Transformation

Therapy guides individuals through a transformative journey of understanding their emotions, thoughts, and needs. It shifts pain into growth, fostering self-compassion and belief in healing. Through this process, one reconnects with themselves, regains hope, and embraces the possibility of change.


*“…At my lowest, the voices in my head told me to end it all. I was lost, terrified, and certain death was my only way out. But beyond the medication, it was the emotional support that brought me back. With time, I began to feel, to see my thoughts and pain not as enemies, but as part of me. This care gave me the strength to face my darkest fears and made me believe that healing is possible, that life can begin again…”*
(P1)


*“…Through psychotherapy, I began to feel a deep shift within. I started to understand my emotions, thoughts, and true needs, and how they were shaped by my past. I realized I could choose healing over pain. I opened myself to love, worth, and acceptance. I learned to protect and care for myself with compassion. I saw the child within me and chose to nurture that part with kindness. I am no longer alone. I am free to grow, to connect, and to believe that change is truly possible…”*
(P5)

#### 3.2.2. Meaningful Relationships Built on Empathy and Mutual Care

This subtheme highlights the transformative power of compassionate relationships in times of struggle. Empathetic connections provide a sense of safety and support, helping individuals face pain and fear, and reaffirming that healing is possible and no one is truly alone.


*“…At my darkest moments, all I needed was one person by my side, willing to listen without judgment and simply be with me. Their empathy and care gave me the strength to face pain and fear, making me feel seen, heard, and truly supported. These relationships became a lifeline, showing me that I wasn’t alone and that I was worthy of healing…”*
(P8)


*“…When I couldn’t trust myself, I borrowed trust from others. My friend didn’t try to fix me—she just stayed. Sometimes she sat with me in silence. That silence meant more than words. It told me that I was worth being with, even at my worst…”*
(P4)

#### 3.2.3. Self-Reconnection Through Understanding, Acceptance, and Compassionate Care

This subtheme reflects the inner process of reconnecting with oneself through gentle awareness and self-acceptance. By turning inward with compassion rather than resistance, participants described finding a quiet strength and emotional stability, allowing them to care for themselves as whole, worthy human beings.


*“…I began to return to myself, to sit with my pain without running from it. I stopped fighting my thoughts and allowed space for them to just be. As I let go of others’ expectations and met myself with kindness, I felt something shift. In that quiet acceptance, I found strength, stillness, and a sense of coming home…”*
(P3)


*“…For a long time, I hated the parts of me that felt too much—too sad, too anxious, too fragile. But the more I tried to suppress those parts, the worse I felt. One day I decided to stop fighting and started listening instead. I realized those parts were trying to protect me. That changed everything. I didn’t need to get rid of them—I needed to care for them…”*
(P6)

### 3.3. Gradually Regaining Agency, Identity, and Meaning in Life

This theme explores the journey of reclaiming agency, rediscovering identity, and finding meaning in life. It focuses on regaining control through decision-making, reconnecting with one’s true self beyond past pain, and finding purpose through growth and meaningful relationships.

#### 3.3.1. Restoring Autonomy and Decision-Making

This subtheme explores the process of reclaiming autonomy through decision-making. It emphasizes how individuals regain control by owning their choices, rebuilding confidence, and realizing their power to shape their future.


*“…For a long time, I felt like I had no control over my life. But as I began making small choices for myself, I started to feel more present, more alive. Each decision reminded me that I still had power, that my voice mattered…”*
(P9)


*“…Even choosing what to eat felt like a big deal in the beginning. I had spent so long letting others decide for me, or just not caring at all. But the day I said, ‘No, I want this,’ I felt something shift. It wasn’t about the food—it was about realizing that I had a say in my life again…”*
(P7)

#### 3.3.2. Rebuilding a Sense of Self by Rediscovering Identity

This subtheme explores the journey of reconnecting with one’s true identity. It emphasizes rediscovering the self beyond pain and external pressures, enabling individuals to rebuild their sense of self and embrace a renewed sense of purpose and confidence.


*“…I had lost myself in trying to meet others’ expectations, but through reflection, I began to reconnect with who I truly was. It wasn’t about creating a new identity, but remembering the person I was before the pain. Rediscovering myself gave me the strength to move forward with purpose…”*
(P7)


*“…For a long time, I only saw myself through the lens of depression—like that was all I was. But slowly, I started remembering the things I loved, the things that made me feel like me. Music, art, just walking in nature. Those weren’t cures, but they reminded me that I was more than my illness. That I still existed beyond the pain…”*
(P3)

#### 3.3.3. Finding Meaning and Purpose in Life Through Growth and Connection

This subtheme highlights how personal growth, combined with meaningful connections, enables individuals to rediscover purpose and direction in life. Through these transformative experiences, they build a deeper sense of identity and find new meaning in their journey.


*“…Through my journey, I began to see that my life could hold meaning beyond the pain I had experienced. I found purpose in my growth, in the connections I built with others, and in the ways, I could contribute to the world around me. It was through these relationships and the personal growth I embraced that I started to feel whole again. My struggles no longer defined me; instead, they became a part of my story, one that has meaning and direction…”*
(P4)


*“…When I started volunteering at the crisis center, I wasn’t just surviving—I was showing others they weren’t alone. My pain taught me to listen and care. I never thought the hardest parts of my life could become meaningful, but they did. Helping others gave my story a purpose I never imagined…”*
(P6)

## 4. Discussion

This study revealed that recovery from severe depression with psychotic symptoms and suicidal behaviors is not a linear process but a deeply transformative and meaning-making journey. The first core theme, understanding and reframing internal experiences, highlights how participants began their recovery by making sense of overwhelming emotional pain and psychotic episodes. Rather than viewing their distress as pathological weakness, they gradually shifted toward interpreting these experiences as meaningful responses to trauma. This transition from self-blame to self-compassion reflects similar processes observed in mindfulness-based and cognitive behavioral therapies, which have demonstrated efficacy in reducing emotional distress and psychotic symptomatology [6,19]. In the Thai cultural context, this reflective process resonates with Buddhist practices of insight (vipassana) and acceptance, which encourage individuals to observe suffering with clarity rather than judgment. Moreover, the growing acceptance of emotional self-exploration among younger Thais adds relevance to this theme within contemporary Thai society [21,22].

Importantly, this study also critiques dominant clinical models that rely on probabilistic suicide risk assessments and symptom focused categorizations. While widely used, such models often overlook the existential and subjective dimensions of suicidality. Baston in Beyond Prediction, argues that suicide risk should not be reduced to statistical likelihood but must be understood in terms of whether suicidal action “belongs to the nearest normal world” of a person, that is, whether suicide appears as a plausible or intelligible option within the individual’s lived reality [27]. The findings of this study support Baston’s view: participants described how recovery involved a shift in their sense of what felt “normal”. Where suicide had once seemed like the only reasonable path, healing involved discovering that life, connection, and hope could become more intelligible and meaningful alternatives. This transformation was not prompted by external assessments, but by a reorientation of lived experience and self-understanding.

The second theme, drawing strength from therapy, relationships, and self-care, underscores the power of relational and emotional support in the recovery process. Participants consistently emphasized that beyond medication, it was the presence of empathic therapists, supportive friends, and the act of caring for themselves that catalyzed healing. These findings align with global research that highlights the importance of psychological interventions in treating psychosis and severe depression, especially when delivered through emotionally safe and supportive relationships [2]. However, in Thailand, systemic stigma and limited access to specialized mental health care remain barriers to such support, especially in rural or underserved areas [3,22]. Many participants reported family relationships that were either distant or conflicted, revealing the need for structured family psychoeducation and community-based support systems. These results emphasize the necessity of integrating culturally appropriate therapeutic alliances and relational healing into Thai mental health care models [16,23]. For example, training psychiatric nurses and community health workers in trauma-informed and relational approaches may improve therapeutic access and reduce stigma in resource-limited settings.

The third theme, gradually regaining agency, identity, and meaning in life, illustrates how participants transitioned from passive survival to active life engagement. Recovery was marked by reclaiming autonomy in decision-making, rediscovering a sense of self beyond illness, and finding renewed life purpose through relationships and contributions. In a collectivist society like Thailand, where social roles and family expectations often define identity, this reclamation of agency is especially significant. For participants who identified as LGBTQ+ or who diverged from traditional gender norms, this process included navigating cultural stigma while rebuilding a positive self-image. International literature has similarly shown that regaining agency and identity is a core mechanism of recovery from psychosis and depression [11,18]. Thai-specific studies also suggest that spiritual growth and social contribution are meaningful sources of recovery, echoing what participants described as “finding purpose again” despite adversity [21,23].

Although all three themes were interdependent, many participants described the reframing of internal experience (Theme 1) as the most psychologically significant and transformative. This internal shift, moving from seeing themselves as broken to viewing their pain as meaningful, often acted as the catalyst for further healing, enabling deeper engagement with therapy, relationships, and life goals. While this study focuses on subjective experiences, it is important to briefly reflect on how these lived transformations relate to clinical understandings of depression. According to DSM 5, major depressive disorder is defined by persistent affective, cognitive, and functional impairments, including hopelessness, psychotic symptoms, and suicidality [26]. Although participants did not speak the diagnostic language, their narratives revealed improvement in core symptom domains, such as reduced suicidal ideation, greater emotional regulation, and improved social functioning. These shifts resonate with literature suggesting that subjective recovery and clinical symptom relief often interact, and that improvements in meaning making, identity, and agency may contribute to or result from symptom reduction [28].

These three themes suggest a multidimensional model of recovery that extends far beyond symptom remission. In the Thai context, recovery must be understood through a socio-cultural lens that incorporates stigma, family obligations, spirituality, and the centrality of interpersonal relationships. Current mental health service delivery in Thailand remains largely hospital centered and pharmacologically driven, with limited integration of recovery oriented or psychosocial interventions [8,22]. Enhancing the roles of psychiatric nurses, peer support workers, and community-based mental health services could improve engagement, promote self-determination, and reduce stigma. Embedding practices that foster narrative reconstruction, identity rebuilding, and empowerment into routine care is essential for supporting long-term recovery in ways that are personally meaningful and culturally responsive [1,19].

## 5. Limitations

While this study offers valuable insights into the lived experiences of recovery from severe depression with psychotic symptoms and suicidal behaviors, several limitations should be acknowledged. First, all participants had achieved stable recovery and were living functional lives at the time of the interview. This approach may have excluded perspectives from individuals who are still struggling, have experienced relapse, or are navigating ongoing instability. Second, the study relied on retrospective accounts, with participants reconstructing emotionally intense and complex periods from memory. As such, the data may be influenced by memory lapses, memory distortions or selective recall. Additionally, while many participants described profound transformation, continuing struggles such as internalized stigma or societal discrimination were still present and may continue to affect long-term recovery. Future research could benefit from longitudinal or real time approaches that capture the dynamics of recovery as they unfold and include more diverse voices, such as individuals currently in crisis or those marginalized by existing care systems.

## 6. Conclusions

This study demonstrates that recovery from severe depression with psychotic symptoms and suicidal behaviors is not only achievable but represents a profound transformation of self and meaning. The lived experiences reveal that healing arises through deep internal understanding, compassionate relationships, and regaining control and identity, which together rebuild hope and purpose. In the Thai cultural context, where stigma and fragmented care often hinder recovery, these findings call for mental health services to embrace holistic, culturally sensitive, and person-centered approaches that honor individuals’ stories and resilience. Mental health nurses and professionals have a critical role in creating safe, empathetic spaces that empower individuals to reclaim agency and rebuild their lives beyond illness. This research provides vital insight and hope, guiding the development of sustainable, recovery-oriented care that moves beyond symptom management toward true healing and social reintegration.

## Figures and Tables

**Table 1 ijerph-22-01606-t001:** Subthemes and themes derived from the data.

Subthemes	Themes
Making Sense of Trauma as a Step Toward Healing	Understanding and Reframing Internal Experiences
Reconstructing Self-Narratives with Compassion	
Therapy as a Journey of Self-Discovery, Healing, and Transformation	Drawing Strength from Therapy, Relationships, and Self-care
Meaningful Relationships Built on Empathy and Mutual Care	
Self-Reconnection through Understanding, Acceptance, and Compassionate Care	
Restoring autonomy and decision-making	Gradually Regaining Agency, Identity, and Meaning in Life
Rebuilding a Sense of Self by Rediscovering Identity	
Finding Meaning and Purpose in Life through Growth and Connection	

## Data Availability

The datasets presented in this article are not publicly accessible due to privacy constraints. To request access to the datasets, please contact Saifon Aekwarangkoon.

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
