# Peer review of "Lived Experiences of Recovery from Severe Depression with Psychotic Symptoms and Suicidal Behaviors: A Phenomenological Study"

_ijerph, 2025, doi:10.3390/ijerph22111606_

Round 1

Reviewer 1 Report

Comments and Suggestions for Authors

The study explores how individuals who experienced depression with psychotic symptoms and suicidal behaviors make sense of their recovery. It focuses on the lived experience of recovery rather than solely on clinical outcomes, aiming to understand what helps patients regain stability, identity, and meaning in life. What helps is reflected in the core themes identified: understanding and reframing internal experiences, drawing strength from therapy and social relations, and regaining agency, identity, and meaning in life. I appreciate the qualitative study and suggest revisions.

For every subtheme, the authors provide only a single quotation. This risks reducing participants’ lived experiences to mere illustrations rather than demonstrating how the themes emerged through engagement with the data. As a reviewer, I expect the authors to establish the credibility of their interpretation by showing how themes were grounded in participants’ accounts. One quotation per subtheme risks looking like cherry-picking. Ideally, two to four quotations per subtheme should be used to highlight both commonalities and contrasts. Without attention to tensions or contradictions, it is difficult to appreciate the richness of description that makes phenomenological research especially valuable.

The study notes that “all participants had achieved stable recovery and were living functional lives.” This is a limitation that should be acknowledged. It excludes those still struggling or experiencing relapse. It would also be important to explore continuing struggles among participants, such as stigma, that may continue to undermine recovery. Furthermore, the accounts are retrospective, with participants recalling struggles and turning points rather than providing self-reports during the transformative journey itself. As such, the data may be affected by memory distortions.

The study implicitly critiques probabilistic, symptom-focused models, noting that these approaches miss the lived experience of suicidality and recovery. To strengthen this critique, the authors could engage with Baston (2024, Beyond Prediction). Baston argues that suicide risk is not a matter of probability but of whether suicidal action belongs to the “nearest normal worlds” for an individual. The study’s findings resonate with this account: recovery involved a shift in participants’ lived sense of normalcy, from suicide as a plausible option to life as the more intelligible and meaningful path.

Author Response

REVIEWER 1

Comments 1: For every subtheme, the authors provide only a single quotation. This risks reducing participants’ lived experiences to mere illustrations rather than demonstrating how the themes emerged through engagement with the data. As a reviewer, I expect the authors to establish the credibility of their interpretation by showing how themes were grounded in participants’ accounts. One quotation per subtheme risks looking like cherry-picking. Ideally, two to four quotations per subtheme should be used to highlight both commonalities and contrasts. Without attention to tensions or contradictions, it is difficult to appreciate the richness of description that makes phenomenological research especially valuable.

Response 1: Thank you for this valuable comment. I have added more quotations for each subtheme to better represent the participants’ lived experiences and to enhance the credibility of the interpretations. Details of these additional quotations can be found on pages 5 to 8 of the revised manuscript.

Comments 2: The study notes that “all participants had achieved stable recovery and were living functional lives.” This is a limitation that should be acknowledged. It excludes those still struggling or experiencing relapse. It would also be important to explore continuing struggles among participants, such as stigma, that may continue to undermine recovery. Furthermore, the accounts are retrospective, with participants recalling struggles and turning points rather than providing self-reports during the transformative journey itself. As such, the data may be affected by memory distortions.

Response 2: Thank you for this valuable comment. In response, I have added a new section titled Limitations as Section 5 on pages 9-10 to acknowledge and address this issue. This addition highlights the retrospective nature of the data, the exclusion of participants who are still struggling or experiencing relapse, ongoing challenges such as stigma that may continue to undermine recovery, and the potential impact of memory distortions.

Comments 3: The study implicitly critiques probabilistic, symptom-focused models, noting that these approaches miss the lived experience of suicidality and recovery. To strengthen this critique, the authors could engage with Baston (2024, Beyond Prediction). Baston argues that suicide risk is not a matter of probability but of whether suicidal action belongs to the “nearest normal worlds” for an individual. The study’s findings resonate with this account: recovery involved a shift in participants’ lived sense of normalcy, from suicide as a plausible option to life as the more intelligible and meaningful path.

Response 3: Thank you for this thoughtful and valuable suggestion. In response, I have revised the second paragraph of the Discussion section to incorporate Baston’s (2024) framework from Beyond Prediction. The new text explicitly critiques probabilistic models of suicide risk and integrates Baston’s concept of the “nearest normal world.” This reference strengthens the theoretical grounding of the study’s findings, which emphasize a shift in participants’ lived sense of what is intelligible and meaningful in relation to suicidality and recovery.

Reviewer 2 Report

Comments and Suggestions for Authors

Remarks for the authors

I found the study both interesting and important. Alongside this, I have noted a few comments below that I think would benefit from further consideration.

Abstract

  1. I suggest revising the sentence to read: “Severe depression with psychotic symptoms and suicidal behaviors, is a critical mental health condition requiring comprehensive care.“

(Alternatively, clarify that the focus is on three distinct elements (Severe depression; psychotic symptoms; suicidal behaviors) and that these three will subsequently be connected and addressed together.)

Introduction

  1. Line 77-78: This element is intrapsychic and thus qualitatively distinct from the other four elements, which all represent external sources of support; the text should therefore be revised to reflect this semantic distinction.
  2. Line 97: You begin with the broader category of “psychiatric disorders” and then focus specifically on severe depression. Then, in line 107, you again refer to general psychiatric problems and once more, narrow down to depression (line 112). I suggest revising the text for greater coherence: first provide the general psychiatric background, and only then narrow the discussion to the specific subject of severe depression.

Materials and methods

  1. General: I recognize that this is a qualitative paper; however, it may strengthen the manuscript to add a DSM-5-TR or ICD-11 reference for the relevant diagnosis. Including this information, at least in the Method section and in the description of the inclusion criteria, would enhance clarity and rigor. Alternatively, the authors may wish to briefly explain their decision not to include such references.
  2. Lines 119-122: Please clarify whether the therapeutic protocol you refer to in these lines is the one you recommend in the Conclusion section, or alternatively, whether it is the protocol commonly used in Thailand that you are suggesting should be changed.
  3. Line 127: please provide an example for such criterion a.
  4. Line 131: I counted six questions not five. Please address this point.
  5. Lines 126, 129, and 146 repeat the term “semi-structured.” I suggest avoiding this redundancy, or at least reducing its frequency.
  6. Line 168: Please clarify the text in the paragraph describing "data triangulation" and “peer review by three independent qualitative researchers".
  7. Line 170: Kindly clarify or illustrate how the strategy of “maximum variation sampling” was operationalized in this study.

Results

  1. General: It will help to see some further elaboration regarding the citations presented. Specifically:

-How were these citations selected from among those representing the theme—was there a defined method, or were they chosen simply as “best samples”?

-How representative are the citations presented in the paper of the wider body of data (e.g., did all six participants refer to all themes and to each of the examples within them, or were there notable divergences within the same theme)?

-Were certain themes experienced by the authors as more salient or dominant than others, in terms of their relative psychological weight?

  1. Lines 181-190: In the current version, the demographic and diagnostic details of the participants (e.g., number of participants, gender, age, diagnosis, and religious affiliation) are presented in the Results section. From a methodological perspective, these data belong more appropriately in the Methods section under Participants. The Results section should focus on the findings that emerged from the qualitative analysis, whereas participant characteristics are considered part of the study design and sampling description. I therefore recommend moving this information to the Methods section and keeping the Results section for the thematic or narrative findings. Alternatively, the authors should provide an explanation for their current preference.
  2. It is very important to add a section for research limitations.
  3. I recognize that this is a qualitative study; however, I believe the discussion would benefit from including at least some comparisons or references to DSM-5 TR or ICD 11 criteria of depression. Ideally, this could involve citing existing research that links subjective improvement and well-being to symptom relief. If such literature is not available, I would encourage at least an informed discussion of this aspect.

Author Response

REVIEWER 2

Comments 1: Abstract: I suggest revising the sentence to read: “Severe depression with psychotic symptoms and suicidal behaviors, is a critical mental health condition requiring comprehensive care. “ (Alternatively, clarify that the focus is on three distinct elements (Severe depression; psychotic symptoms; suicidal behaviors) and that these three will subsequently be connected and addressed together.)

Response 1: Thank you for the helpful suggestion. I have revised the sentence in the first two lines of the abstract as recommended. It now reads: Severe depression with psychotic symptoms and suicidal behaviors, is a critical mental health condition requiring comprehensive care.”

Comments 2: Introduction

-Line 77-78: This element is intrapsychic and thus qualitatively distinct from the other four elements, which all represent external sources of support; the text should therefore be revised to reflect this semantic distinction.

-Line 97: You begin with the broader category of “psychiatric disorders” and then focus specifically on severe depression. Then, in line 107, you again refer to general psychiatric problems and once more, narrow down to depression (line 112). I suggest revising the text for greater coherence: first provide the general psychiatric background, and only then narrow the discussion to the specific subject of severe depression.

Response 2: Thank you for the valuable feedback. I have revised the introduction accordingly. Specifically, in the fourth paragraph and under the section Mental health care in the Thai context (page 3), I clarified the intrapsychic nature of the element mentioned in lines 77-78, distinguishing it from the external sources of support. I also improved the coherence and flow by restructuring the discussion to first present the general psychiatric background before narrowing down to severe depression, as suggested.

Comments 3: Materials and methods

-General: I recognize that this is a qualitative paper; however, it may strengthen the manuscript to add a DSM-5-TR or ICD-11 reference for the relevant diagnosis. Including this information, at least in the Method section and in the description of the inclusion criteria, would enhance clarity and rigor. Alternatively, the authors may wish to briefly explain their decision not to include such references.

-Lines 119-122: Please clarify whether the therapeutic protocol you refer to in these lines is the one you recommend in the Conclusion section, or alternatively, whether it is the protocol commonly used in Thailand that you are suggesting should be changed.

-Line 127: please provide an example for such criterion a.

-Line 131: I counted six questions not five. Please address this point.

-Lines 126, 129, and 146 repeat the term “semi-structured.” I suggest avoiding this redundancy, or at least reducing its frequency.

-Line 168: Please clarify the text in the paragraph describing "data triangulation" and “peer review by three independent qualitative researchers".

-Line 170: Kindly clarify or illustrate how the strategy of “maximum variation sampling” was operationalized in this study.

Response 3: We have revised the Materials and Methods section accordingly, specifically in paragraphs 1 to 4 and paragraph 6 on pages 3-4.

-We have added a reference to DSM-5-TR to clarify the diagnostic criteria used for participant inclusion, thereby enhancing the rigor and clarity of the study. Regarding the therapeutic protocol, we clarified that it refers to the protocol commonly used in Thailand, which we suggest should be reconsidered or improved, rather than the one recommended in the conclusion section.
-An example for criterion “a” has been added to provide clearer guidance.
-We reviewed the questions and corrected the number from five to six.
-To avoid redundancy, we reduced the repeated use of the term “semi-structured.”
-The paragraph describing “data triangulation” and “peer review by three independent qualitative researchers” has been revised to clarify these procedures.
-Finally, we elaborated on how the “maximum variation sampling” strategy was operationalized in this study, including specific examples to illustrate its application.

Comments 4: Results

General: It will help to see some further elaboration regarding the citations presented. Specifically:

-How were these citations selected from among those representing the theme—was there a defined method, or were they chosen simply as “best samples”?

-How representative are the citations presented in the paper of the wider body of data (e.g., did all six participants refer to all themes and to each of the examples within them, or were there notable divergences within the same theme)?

-Were certain themes experienced by the authors as more salient or dominant than others, in terms of their relative psychological weight?

Response 4: We have revised the Results section, specifically paragraph 1 on page 5, by adding the following clarification: “The findings highlight the lived experiences of participants as they overcame depression, suicidal tendencies, and psychiatric symptoms, revealing three main themes: (1) understanding and reframing internal experiences, (2) drawing strength from therapy, relationships, and self-care, and (3) gradually regaining agency, identity, and meaning in life. Representative quotations were purposively selected to illustrate these themes, reflecting common patterns across all nine participants. While not every participant explicitly discussed every sub-theme, all themes were broadly shared, with Theme 2 emerging as the most salient, frequently described as a pivotal source of psychological strength. Themes 1 and 3 were also prominent but varied in emphasis among individuals. A comprehensive summary of themes and sub-themes is presented in Table 1.”

Comments 5: Results

- Lines 181-190: In the current version, the demographic and diagnostic details of the participants (e.g., number of participants, gender, age, diagnosis, and religious affiliation) are presented in the Results section. From a methodological perspective, these data belong more appropriately in the Methods section under Participants. The Results section should focus on the findings that emerged from the qualitative analysis, whereas participant characteristics are considered part of the study design and sampling description. I therefore recommend moving this information to the Methods section and keeping the Results section for the thematic or narrative findings. Alternatively, the authors should provide an explanation for their current preference.

Response 5:  We have revised the manuscript by moving the demographic and diagnostic details of the participants to the Materials and Methods section, specifically in paragraph 2 (lines 133-139) on pages 3-4. This adjustment aligns with methodological standards by placing participant characteristics under the study design and sampling description, while keeping the Results section focused on the thematic findings.

Comments 6: It is very important to add a section for research limitations.

Response 6: We have added a new section titled Limitations on pages 9–10 to address the research limitations.

Comments 7: I recognize that this is a qualitative study; however, I believe the discussion would benefit from including at least some comparisons or references to DSM-5 TR or ICD 11 criteria of depression. Ideally, this could involve citing existing research that links subjective improvement and well-being to symptom relief. If such literature is not available, I would encourage at least an informed discussion of this aspect.

Response 7: We have addressed this suggestion by including comparisons and references to DSM-5 TR and ICD-11 criteria for depression, as well as relevant literature linking subjective improvement and well-being to symptom relief, in paragraph 5 on page 9. Corresponding references have also been added to the reference list.

Round 2

Reviewer 1 Report

Comments and Suggestions for Authors

I think the authors did an excellent job with revising their manuscript, and I suggest acceptance as it is.